# Nicotinamide Phosphoribosyltransferase (Nampt)/Nicotinamide Adenine Dinucleotide (NAD) Axis Suppresses Atrial Fibrillation by Modulating the Calcium Handling Pathway

**DOI:** 10.3390/ijms21134655

**Published:** 2020-06-30

**Authors:** Duo Feng, DongZhu Xu, Nobuyuki Murakoshi, Kazuko Tajiri, Rujie Qin, Saori Yonebayashi, Yuta Okabe, Siqi Li, Zixun Yuan, Kazutaka Aonuma, Masaki Ieda

**Affiliations:** Department of Cardiology, Institute of Clinical Medicine, Faculty of Medicine, University of Tsukuba, Tsukuba, Ibaraki 305-8575, Japan; fengduoryu@outlook.com (D.F.); n.murakoshi@md.tsukuba.ac.jp (N.M.); ktajiri@md.tsukuba.ac.jp (K.T.); leopalace@sohu.com (R.Q.); syonebayashi789@gmail.com (S.Y.); yokabe0211@gmail.com (Y.O.); l.siqi@outlook.com (S.L.); YUAN1120364808@yahoo.com (Z.Y.); kaonuma@md.tsukuba.ac.jp (K.A.); mieda@md.tsukuba.ac.jp (M.I.)

**Keywords:** Nampt, NAD, AF, calcium handling, ROS, CaMKII, RyR2, cardiac myocytes, Sirt1

## Abstract

Aging and obesity are the most prominent risk factors for onset of atrial fibrillation (AF). Nicotinamide phosphoribosyltransferase (Nampt) is the rate-limiting enzyme that catalyzes nicotinamide adenine dinucleotide (NAD) activity. Nampt and NAD are essential for maintenance of cellular redox homeostasis and modulation of cellular metabolism, and their expression levels decrease with aging and obesity. However, a role for Nampt in AF is unknown. The present study aims to test whether there is a role of Nampt/NAD axis in the pathogenesis of obesity-induced AF. Male C57BL/6J (WT) mice and heterozygous Nampt knockout (NKO) mice were fed with a normal chow diet (ND) or a high-fat diet (HFD). Electrophysiological study showed that AF inducibility was significantly increased in WT+HFD, NKO+ND, and NKO+HFD mice compared with WT+ND mice. AF duration was significantly longer in WT+HFD and NKO+ND mice and further prolonged in NKO+HFD mice compared with WT+ND mice and the calcium handling pathway was altered on molecular level. Also, treatment with nicotinamide riboside, a NAD precursor, partially restored the HFD-induced AF perpetuation. Overall, this work demonstrates that partially deletion of Nampt facilitated HFD-induced AF through increased diastolic calcium leaks. The Nampt/NAD axis may be a potent therapeutic target for AF.

## 1. Introduction

Atrial fibrillation (AF) is the most common arrhythmia encountered in clinical practice and is characterized by irregular atrial electrical activity resulting in asynchronous atrial contraction. AF increases the risk of embolic stroke, heart failure, and overall mortality [1,2,3], and is a global problem affecting more than 33 million people worldwide [4]. Previous studies showed that aging and obesity are two of the most prominent risk factors for AF occurrence [5,6,7]. In obesity, cardiometabolic changes in the heart favor reactive oxygen species (ROS) production [8,9]. Oxidative stress activates calmodulin-dependent protein kinase II (CaMKII), leading to arrhythmogenic hyperphosphorylation of RyR2. Oxidized CaMKII-mediated hyperphosphorylation of RyR2 promotes diastolic calcium leaks from the sarcoplasmic reticulum (SR) in atrial cardiomyocytes, causing AF [10,11]. Therefore, oxidized CaMKII and its downstream components may be promising therapeutic targets for prevention of AF.

Nicotinamide phosphoribosyltransferase (Nampt) is a regulator of the intracellular nicotinamide adenine dinucleotide (NAD) pool [12]. NAD is an essential coenzyme involved in cellular senescence and redox reactions and is a substrate for NAD-dependent enzymes like sirtuin 1 (Sirt1) [13]. The Nampt/NAD/Sirt1 axis has protective roles against aging and obesity-related disorders such as type 2 diabetes mellitus by influencing the oxidative stress response, apoptosis, lipid and glucose metabolism, inflammation, and insulin resistance [14,15,16,17,18]. Nampt is highly expressed in cardiomyocytes, and recent studies have demonstrated its effects on a variety of cardiac morbidities, including dilated cardiomyopathy, ischemia/reperfusion injury, and heart failure [19,20,21]. However, there have been no reports on the relationship between AF and Nampt. Because the Nampt/NAD axis is involved in obesity and aging, the most important risk factors for AF, we hypothesized that the Nampt/NAD axis has an important role in the pathogenesis of obesity-induced AF. To achive that, we used heterozygous Nampt knockout mice (NKO), wild-type (WT) littermate mice as control and tried to clarify whether Nampt is involved in an 8 weeks high-fat-diet (HFD)-induced atrial arrhythmogenic remodeling mice.

## 2. Results

### 2.1. HFD Reduces Nampt Expression in the Atrial Tissues

First, we confirmed the expression of Nampt in the atrial tissues. From 5 weeks of age, NKO and WT mice were fed with either a standard chow (normal diet; ND) or 60% HFD for 8 weeks. Thus, there were four groups of mice: WT+ND mice, WT+HFD mice, NKO+ND mice and NKO+HFD mice. As shown in Figure 1A, real-time PCR demonstrated that the Nampt mRNA expression level was decreased by approximately 50% in WT+HFD and NKO+ND mice compared with WT+ND mice. The Nampt downstream target Sirt1 was also downregulated by approximately 50% under the influence of NKO and HFD (Figure 1B). And the nicotinamide riboside kinase 2 (Nmrk2), another NAD precursor from the nicotinamide riboside (NR) pathway was significantly increased in NKO+HFD group compared to the WT+ND group and the tendency of increase in NKO+ND and WT+HFD groups but not statistical significance (Figure 1C). Western blotting of Nampt and Sirt1 revealed significantly downregulated protein expressions like the qPCR results (Figure 1D–F). On immunohistochemistry, Nampt was mainly expressed in the cytoplasm of atrial cardiomyocytes as indicated in Figure 1G. The immunohistochemical intensity was also reduced by nearly 50% in NKO+ND and WT+HFD mice compared with WT+ND mice, and were further decreased in NKO+HFD mice (Figure 1G,H). Finally, we evaluated the NAD level in the atrial tissues, as shown in Figure 1I, there were significant decreases of NAD/NADH ratio in other three groups compared to the WT+ND group. Taken together, Nampt was mainly distributed in the cytoplasm of atrial cardiomyocytes and was significantly decreased by HFD and NKO, and Nampt downstream proteins NAD and Sirt1 were both decreased due to the Nampt loss despite the attempt of Nmrk2 pathway to restore the NAD level.

### 2.2. NKO and HFD Do Not Affect Cardiac Function and Morphology

To assess cardiac function and morphology, we performed echocardiographic analysis (Figure 2A and Table 1). The analysis revealed that neither NKO nor HFD had any effects on cardiac function nor morphology. (Table 1). We further examined the fibrotic area and cardiomyocytes size of the atrial tissues by Masson trichrome staining (Figure 2B). The results showed no significant difference in fibrotic area and atrial cardiomyocytes width among the four experimental conditions with a group size of 5 mice (Figure 2C,D).

### 2.3. HFD Increases Body Weight (BW) and Fat Volume, but Not Affected by NKO

Although Nampt is an adipokine with an important role in energy metabolism and adipose tissue formation, heterozygous NKO did not affect BW in mice fed the ND or HFD, while HFD feeding significantly increased BW in both WT and NKO mice as expected (Table 2). Blood pressure was only significantly increased in NKO+HFD mice. Furthermore, the ratio of heart weight to BW was significantly increased in HFD-fed groups (Table 2). We also performed CT scans to evaluate the amounts of subcutaneous fat and visceral fat (Figure 2D). As expected, HFD feeding significantly increased both subcutaneous fat and visceral fat. However, heterozygous NKO did not affect the amount or distribution of either fat (Figure 2E,F).

### 2.4. NKO and HFD Increase AF Susceptibility and Duration

To evaluate AF vulnerability, atrial burst pacing was carried out through a transvenous electrode catheter. After 30 s of burst pacing, AF could be induced and terminated spontaneously (Figure 3A). AF susceptibility, which was calculated by the percentage of occurrences of the AF after 5 times of burst pacing was significantly increased in WT+HFD, NKO+ND, and NKO+HFD mice compared with WT+ND mice, while no significant difference was observed among ND+HFD, NKO+ND, and NKO+HFD mice (Figure 3B). AF duration was significantly longer in WT+HFD and NKO+ND mice and further prolonged in NKO+HFD mice compared with WT+ND mice (Figure 3C). We also investigated the electrophysiological properties in the four groups. Atrial muscle effective refractory period (AERP) at the basic cycle length of 150 ms was significantly shorter in WT+HFD, NKO+ND, and NKO+HFD mice than in WT+ND mice (Figure 3D). The electrophysiological study indicated that NKO and HFD increased AF persistency with shortening of AERP.

### 2.5. NKO and HFD Accelerate Diastolic Calcium Leaks Under Isoproterenol (Iso) Stimulation in Cardiomyocytes

To evaluate the cellular electrophysiology, we performed calcium imaging of isolated cardiomyocytes from the four groups. Figure 4A shows representative confocal microscopy line-scan images of cardiomyocytes to evaluate Ca^2+^ sparks with or without Iso stimulation. By adding Iso, the calcium releases would be significantly accelerated, with more intensity compared with the baseline condition. The representative images of 3D plot of the calcium sparks are shown in Figure 4B. The Ca^2+^ spark frequency was significantly increased in WT+HFD, NKO+ND, and NKO+HFD mice compared with WT+ND mice under Iso stimulation (Figure 4C). The fractional fluorescence increases (F/F_0_) was significantly increased in WT+HFD, NKO+ND, and NKO+HFD mice compared with WT+ND mice (Figure 4D). The time to peak was shortened in NKO+ND and WT+HFD mice and further shortened in NKO+HFD mice compared with WT+ND mice after Iso stimulation (Figure 4E). Calcium imaging indicated that HFD and NKO both accelerated diastolic Ca^2+^ leaks from the SR.

### 2.6. NKO and HFD Promote CaMKII Oxidation and RyR2 Phosphorylation

Oxidized CaMKII is closely linked to AF, and CaMKII-dependent phosphorylation of RyR2 at Serine-2814 leads to SR calcium leaks that likely contribute to AF. Therefore, we performed western blotting to evaluate oxidized CaMKII and phosphorylated RyR2. As shown in Figure 5A, NKO+HFD mice had significantly elevated expression of oxidized CaMKII, despite no significant difference in total CaMKII among the four groups (Figure 5B). RyR2 phosphorylated at Ser-2814 was also significantly increased in NKO+HFD mice compared with WT+ND mice (Figure 5C). These data suggest that NKO and HFD promoted oxidized CaMKII-dependent RyR2 phosphorylation.

### 2.7. NAD Precursor NR Increases Nampt Amount and Decreases AF Duration

Finally, to assess whether NAD precursor NR can abolish HFD-induced arrhythmogenesis, we compared ND, HFD and HFD mice treated with NR. Western blotting analysis showed that treatment with NR recovered the reduction in Nampt expression in HFD mice (Figure 6A,B). The NAD levels were also recuperated by the NR treatment (Figure 6C). The electrophysiological study revealed that, despite the lack of difference in AF inducibility, treatment with NR restored HFD-induced AF persistency and shortened AERP (Figure 6D–F).

## 3. Discussion

In this study, HFD feeding and heterozygous NKO prolonged catheter-induced AF duration without significant structural abnormalities. HFD and NKO enhanced the expressions of oxidized CaMKII and phosphorylated RyR2 and induced frequent diastolic Ca^2+^ leaks in cardiomyocytes. Furthermore, NAD precursor NR partially rescued the arrhythmogenic phenotypes. It is suggested that the Nampt/NAD axis plays a protective role in AF pathogenesis by regulating the calcium handling pathway.

Nampt is the rate-limiting enzyme in the salvage pathway for NAD biosynthesis, which regulates Sirt1 activity. Nampt and NAD levels in the liver and adipose tissue were decreased by HFD intake [16,22]. In the present study, we confirmed that the Nampt, NAD and Sirt1 levels were significantly reduced in the atrial tissues of Nampt knockout mice and mice fed the HFD. Moreover, we found that BW and adipose tissue weight did not differ significantly between WT+HFD and NKO+HFD mice. A previous study showed that adipose tissue-specific NKO mice were resistant to HFD-induced obesity [23]. This discrepancy may have arisen through the differences between heterozygous conventional knockout mice and adipose tissue-specific homozygous knockout mice.

The electrophysiological study demonstrated that both NKO and HFD significantly decreased AERP and significantly increased AF susceptibility and duration, indicating that electrical remodeling occurred with NKO and HFD. Electrical remodeling is known to be a main mechanism for AF pathogenesis, characterized by qualitative and quantitative alterations to ion channels, pumps, and exchangers [24]. A previous study clarified the relationship between oxidized CaMKII and AF pathogenesis [10]. Oxidized CaMKII enhanced phosphorylation of RyR2 at Serine-2814, thereby facilitating diastolic calcium leaks from the SR that trigger AF [11,25]. In the present study, we found that oxidized CaMKII and phosphorylated RyR2 were significantly increased in NKO+HFD mice. NAD is phosphorylated by NAD kinase to form NADP, which is subsequently reduced to NADPH by NADP dehydrogenase. NADPH acts as an antioxidant to neutralize the high levels of ROS generated by increased metabolic activity [26]. Moreover, Nampt was shown to suppress oxidative stress by increasing NADPH levels in cell [27,28], while NAD precursor NR supplementation partially restored diabetic-induced degradation of NADH and NADPH levels [29]. Meanwhile, NADPH oxidase has emerged as an important enzymatic source for ROS production in AF [30,31]. Therefore, the present findings suggest that HFD and NKO-induced NADPH degradation increased ROS production, which directly enhanced oxidation of CaMKII and phosphorylation of RyR2, thereby augmenting AF occurrence. Moreover, we clearly demonstrated by calcium imaging that Ca^2+^ sparks and mini waves were increased by either NKO or HFD, and further increased in NKO+HFD mice. The frequencies of Ca^2+^ sparks and mini waves measured by confocal microscopy are directly correlated with diastolic SR Ca^2+^ leaks [11]. Therefore, it is strongly suggested that both NKO and HFD facilitate diastolic Ca^2+^ leaks from the SR. Furthermore, the kinetics of the calcium leaks were altered by the influences of NKO and HFD. F/F_0_, an indicator of calcium spark amplitude, was significantly increased, while time to peak, an indicator of calcium leak propagation speed, was significantly shortened in both NKO and HFD mice, and further shortened in NKO+HFD mice. Taken together, we conclude that NKO and HFD facilitated diastolic calcium leaks from the SR through enhanced oxidized CaMKII-dependent hyperphosphorylated RyR2, leading to augmentation of AF persistence.

Structural remodeling is another mechanism characterized by atrial enlargement and tissue fibrosis, as key determinants for AF-maintaining reentry [24]. Previous studies showed that the Nampt/NAD/Sirt1 axis has a protective effect in several heart diseases through elimination of oxidative stress, including ischemic heart disease, cardiomyopathy, pressure-overload heart failure, and cardiac hypertrophy [21,32,33,34]. Moreover, Nampt was reported to directly regulate apoptosis in cardiomyocytes [35]. Although cardiac morphology and function remained unchanged in HFD and NKO mice in the present study, it is possible that the Nampt/NAD/Sirt1 axis may have had a positive impact on modulation of structural remodeling and/or cardiac metabolism, leading to suppression of AF vulnerability.

Previous studies suggested that inflammation is closely linked to AF pathogenesis [36] through fibroblast activation and extracellular matrix remodeling under the influences of inflammatory cytokines like tissue necrosis factor-*α* and interleukin-6. Meanwhile, Nampt was recently reported to modulate inflammatory reactions by reducing NF-κB production in adipose tissue [37]. Therefore, Nampt may have a role in AF pathogenesis through modulation of inflammation. Whether the Nampt/NAD axis regulates AF occurrence through modulation of inflammation remains to be elucidated.

Previous studies had shown that NAD can be synthesized by multiple pathways and key enzymes such as nicotinamide riboside kinase (Nmrk 1,2) [38] as well as NAD synthetase (Nadsyn1) [39]. And under the Nampt deficient condition, the Nmrk2 pathway was activated but could not recuperate the NAD loss.

Finally, we used NR, a pyridine-nucleoside form of vitamin B that functions as a NAD precursor [40], to assess whether NR can recover the negative effects arising from NKO. NR has been proven to effectively increase NAD synthesis in various tissues and organs such as the heart, skeletal muscle, adipose tissue, and liver [19,29,41,42]. Our data showed that NR supplementation significantly increased Nampt and NAD levels in HFD-fed mice. Also, AF duration and AERP were both significantly recovered by NR supplementation. Although it is not clear why NR treatment could recover HFD-induced AF prolongation and AERP shortening, NR treatment may improve electrical remodeling through modulating calcium-handling pathway. Taking our data and previous findings together, NR may be a promising therapeutic agent for cardiovascular diseases, including AF, in clinical practice.

Even though we did not test the ROS production, there have been multiple studies showing that the decrease of the Nampt expression leads to NAD reduction, and increase of the ROS production in several cells and tissues [34,43,44]. Further works will be required to determine the NR treatment effects on calcium images. Also, Pharmacokinetics and pharmacodynamics of therapeutic NR administration in animal models and people with AF are needed to enable safe and effective human translation.

In conclusion, we found that heterozygous NKO directly enhanced oxidized CaMKII-mediated phosphorylation of RyR2, and diastolic Ca^2+^ leaks from the SR in cardiomyocytes, leading to facilitation of AF in HFD-fed mice. Supplementation with NAD precursor NR partially rescued the high AF vulnerability. Therefore, the Nampt/NAD axis may be a novel therapeutic target for AF treatment.

## 4. Materials and Methods

### 4.1. Mouse Model

After receiving approval from the Institutional Animal Experiment Committee of the University of Tsukuba (approved number: 140281 approved date: 7 April 2016), animal experiments were carried out in accordance with the Guide for the Care and Use of Laboratory Animals published by the US National Institutes of Health, the Regulation for Animal Experiments in the University of Tsukuba, and the Fundamental Guideline for Proper Conduct of Animal Experiments and Related Activities in Academic Research Institutions under the jurisdiction of the Ministry of Education, Culture, Sports, Science and Technology of Japan.

Systemic Nampt heterozygous knockout mice with a C57BL/6 genetic background were generalized and genotyped as described previously [45], and WT littermate mice were used as controls. From 5 weeks of age, NKO and WT male mice were fed with either standard chow (normal diet; ND) or 60% HFD for 8 weeks. After 8 weeks on the different diets, the mice were examined and then sacrificed for further analysis.

### 4.2. Blood Pressure Measurement

Blood pressure measurements were performed in conscious mice via the tail-cuff CODA non-invasive blood pressure system (KENT Scientific Corporation, Torrington, CT, USA).

### 4.3. Echocardiography

Echocardiography was conducted as described previously [40,46]. At 5 and 13 weeks of age, parasternal long-axis and short-axis views were obtained at the papillary muscle level under anesthesia with isoflurane using an echocardiographic system (Vevo 2100; Visual Sonics, Toronto, Canada). The heart rate (HR), left ventricular end-diastolic diameter (LVDd), left ventricular end-systolic diameter (LVD’s), fractional shortening (FS), left ventricular ejection fraction (LVEF), left atrial dimension (LAD), and left ventricular systolic volume (LVV) were determined. LVEF was calculated by the Teichholz method.

### 4.4. Body Fat Composition Analysis

Body fat composition was analyzed as previously described [47] with some modifications. Briefly, mice were anesthetized by inhalation of isoflurane and scanned using a LaTheta LCT-100 Experimental Animal X-ray CT System (Hitachi-Aloka Medical, Tokyo, Japan). Contiguous 1-mm slice images between the proximal end of lumbar vertebra L1 and the distal end of lumbar vertebra L6 were quantitatively assessed using LaTheta version 2.10 software (Hitachi-Aloka Medical). Subcutaneous fat and visceral fat were quantified separately.

### 4.5. Electrophysiological Study and AF Induction

The electrophysiological study and AF induction were performed as previously described with some minor modifications [48,49]. Briefly, a right cervical vein cut was performed, and a 2-Fr quadripolar electrode catheter with 2-mm electrode distance (Unique Medical, Tokyo, Japan) was inserted into the right atrium to perform the electrophysiological study. The catheter was placed at the site where the amplitude of the intraatrial electrogram was higher than that of the intraventricular electrogram. To measure AERP, a programmable stimulator (SEN-7203; Nihon Kohden, Tokyo, Japan) was used to deliver approximately twice the threshold current at 2-ms duration. AERP was measured at basic cycle lengths involving a train of eight basic stimuli (S1 × 8) followed by a single extra stimulus (S2) at 5-ms decrements. AERP was defined as the longest S1–S2 interval that failed to be captured.

For AF induction, atrial burst pacing was delivered through two poles on the electrode catheter by a programmable stimulator with amplitude of 6 V, cycle length of 20 ms, pulse duration of 6 ms, and stimulation time of 30 s. Atrial burst pacing was performed five times consecutively in each mouse to obtain inducibility. AF duration was defined as the interval between initiation and spontaneous termination of AF.

### 4.6. Calcium Imaging

Adult atrial cardiomyocytes were isolated from WT and NKO mice atriums using a Langendorff-free procedure [50]. The isolated cells were incubated with Fluo-4 AM (Invitrogen, Carlsbad, CA, USA) diluted in 20% Pluronic F-127 DMSO at 5 µM final concentration in Tyrode buffer (NaCl 140 mM, KCl 5 mM, HEPES 5 mM, NaH_2_PO_4_ 2 mM, MgCl_2_ 1 mM, CaCl_2_ 2 mM, glucose 10 mM, pH 7.4) for 10 min. The cells were then rinsed with Tyrode buffer and maintained in the buffer during experiments. To capture the Fluo-4 AM signals, live cell imaging was performed with a 40× lens on a TCS SP5 Confocal Microscope System (Leica Microsystems, Wetzlar, Germany). Line scan images were recorded along the longitudinal axis of the cell at 500 Hz, with a pixel size of 100 nm and pinhole optimized for a resolution of 0.4 μm in the focal plane and <1 μm in the z-axis. Iso (100 nM) was added to the cell after the baseline stablished. Baseline fluorescence (F_0_) was determined by averaging 10 images without calcium spark activity. Fractional fluorescence increases (F/F_0_) were determined in areas (2.2 × 2.2 μm) where calcium sparks were detected. Calcium sparks were defined as local fractional fluorescence increases >1.3. Calcium sparks were detected and characterized following established criteria [51]. To evaluate calcium sparks, cytoplasmic fluorescence signals were obtained using LAS-X software (Leica Microsystems) and data were analyzed by LAS-X software and ImageJ version 1.45 software.

### 4.7. RNA Expression Analysis

Briefly, total RNA was extracted from the atrial tissues using an RNeasy Fibrous Tissue Mini Kit (Qiagen, Venlo, The Netherlands), and 2 µg of total RNA was reverse-transcribed to cDNA with a High-Capacity cDNA Reverse Transcription Kit (Thermo Fisher Scientific Inc., Waltham, MA, USA). The mRNA expression levels of the target genes were analyzed by an Applied Biosystems 7500 Real-Time PCR System or QuantStudio 5 (Thermo Fisher Scientific). The commercially available gene-specific primers and probe sets used were obtained from Integrated DNA Technologies (Coralville, IA, USA). The PCR mixture (10 µL total volume) contained the primer and probe for each gene at 250 nM, and PrimeTime Gene Expression Master Mix (Integrated DNA Technologies). PCR amplification was performed in duplicate as follows: 1 cycle at 95 °C for 10 min, followed by 40 cycles at 94 °C for 15 s and 60 °C for 1 min. To ensure the desired product was amplified, melt-curve analysis was used to determine that only a single peak was present representing a single PCR product, and the amplification curve of a two-fold dilution series of the cDNA was analyzed to determine that the primer pairs were not amplifying primer-dimers, as per MIQE guideline [52]. The quantitative values of target mRNA were normalized to the expression of 18S rRNA (4319413E, Thermo Fisher Scientific). The following primers were used: Sirt1: Mm. 00490758_m1 (ThermoFisher); Ryr2: Mm.PT.45974879; Nampt: Mm.PT.58.10977319; Camk2a: Mm.PT.58.8246010; Nmrk2: Mm.PT.58.15829037 (Integrated DNA Technologies, Coralville, IA, USA).

### 4.8. Histology and Immunohistochemistry

Tissues were fixed with 4% paraformaldehyde, embedded in paraffin, sectioned into 4-µm-thick slices, and stained with Masson’s trichrome. Images were obtained with a digital microscope (Biozero BZ-X700; Keyence, Osaka, Japan), and collagen deposition was quantified by dividing the collagen deposition area by the total area. The analysis was performed using ImageJ version 1.45 software (NIH, Bethesda, MD, USA).

To evaluate Nampt expression, we performed immunohistochemistry. After deparaffinization and antigen activation, the sections were incubated with a rabbit anti-Nampt monoclonal antibody (LS-C137911; LifeSpan BioSciences, Seattle, WA, USA) at 4 °C overnight. Subsequently, the sections were thoroughly rinsed with PBS and incubated with horseradish peroxidase (HRP)-conjugated goat anti-rabbit IgG (ab6721; Abcam, Cambridge, UK) at room temperature for 1 h. After rinsing with PBS and mounting on slides with mounting medium containing DAPI, images were captured with the Biozero BZ-X700 digital microscope. The ratio of positive staining area to total area was calculated using ImageJ version 1.45 software.

### 4.9. Western Blotting Analysis

Western blotting was performed as described previously [40,45]. Briefly, isolated atrial tissues were homogenized in PRO-PREP protein extraction solution (iNtRON Biotechnology Inc. Kyungki-Do, Korea), and the supernatants were used for western immunoblotting. Appropriate volumes of the samples (10 µg protein) were mixed with an equal volume of sample buffer, heated at 95 °C for 5 min, and subjected to SDS-PAGE using polyacrylamide gels (Bio-Rad Laboratory, Hercules, CA, USA). The proteins were transferred from the gels to polyvinylidene difluoride membranes by semidry electroblotting. The resulting membranes were incubated at room temperature for 1 h with the following primary antibodies: anti-Nampt (11776-1-AP; LifeSpan BioSciences); anti-SIRT1 (P-20) (sc-19857; Santa Cruz Biotechnology, Santa Cruz, CA, USA); anti-Serca2 (2A7-A1) (MA3-919; Thermo Fisher Scientific Inc.); anti-phospho-Serca2 (A010-25AP; Badrilla, Leeds, UK); anti-RyR2 (pSer2814) (A010-31AP; Badrilla); anti-RyR2 (PA5-36121; Thermo Fisher Scientific); anti-CaMKII (07-1496; Merck Millipore, Billerica, MA, USA); anti-oxidized CaMKII (T286) (07-1387; Merck Millipore); and anti-β-actin (4967; Cell Signaling Technology, Temecula, CA, USA). The membranes were then incubated with an appropriate second antibody at 4 °C for another one hour, HRP-conjugated goat anti-rabbit IgG (ab6721; Abcam) or HRP-conjugated rabbit anti-mouse IgG (ab97046, Abcam). Immunoreactions were visualized with an enhanced chemiluminescence method (ECL Prime Western Blotting Detection; GE Healthcare, Hertfordshire, UK). Densitometric analysis was performed on the scanned immunoblot images from 2–3 independent measurements (3 mice per group for each measurement) using ImageJ version 1.45 software to evaluate the ratios of target protein to β-actin or phosphorylated form to total protein.

### 4.10. NAD Detection Assay

NAD/NADH ratio was determined using NAD/NADH Quantification Kit (MAK037; Sigma-Aldrich, St. Louis, MO, USA). Four atrial tissues in each group were homogenized in NADH/NAD extraction buffer. The samples were clarified by centrifugation, and the supernatant was deproteinized by filtering it though a 10-kDa spin column (ab93349, Abcam). The assay was then performed according to the manufacturer’s instructions. Values were corrected for dilutions and protein content of the samples.

### 4.11. NR Treatment

Wild-type mice were started to be fed with either chow diet or HFD diet at the age of 5 weeks old and were injected intraperitoneally with same amount of either saline solution (WT and HFD group) or saline+NR (300 mg/kg body weight) with daily basis for 8 weeks. Then further studies were conducted afterwards.

### 4.12. Statistical Analysis

All data are expressed as mean ± standard error of the mean. For comparisons between the groups, continuous values were analyzed by one-way ANOVA followed by a post-hoc Bonferroni test. The results for each experiment were obtained from three independent measurements. Significance was accepted for values of *p <* 0.05. All statistical analyses were performed using IBM SPSS version 21.0 statistical software (IBM Co. Ltd., Armonk, NY, USA).

## Figures and Tables

**Figure 1 ijms-21-04655-f001:**
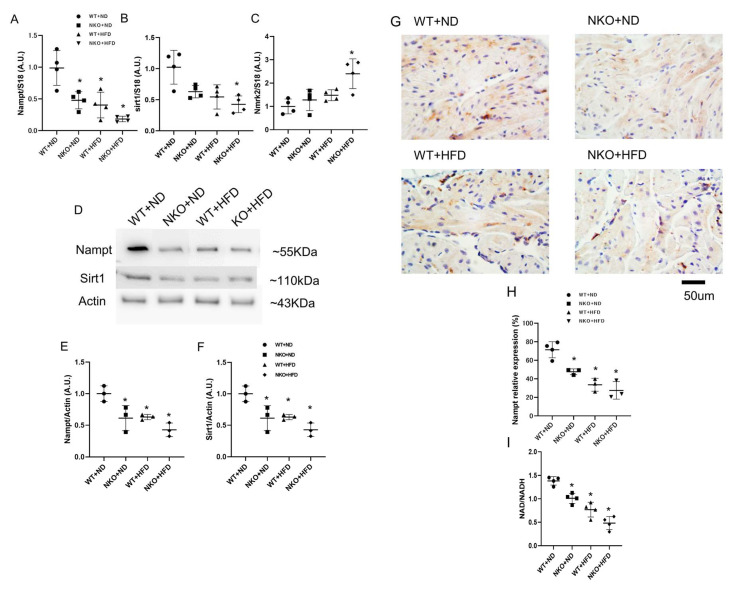
Gene and protein expressions of Nampt and Sirt1 in the atrial tissues. (**A**) Nampt mRNA expression levels in the atrial tissues evaluated by quantitative PCR (*n* = 4 mice per each group). (**B**) Sirt1 mRNA expression levels in the atrial tissues evaluated by quantitative PCR (*n* = 4 mice per group). (**C**) Nmrk2 mRNA expression levels in the atrial tissues evaluated by quantitative PCR (*n* = 4 mice per group). (**D**) Representative western blots for Nampt and Sirt1 in the atrial tissues (*n* = 3 mice per group). (E) Nampt protein expression levels in the atrial tissues (*n* = 3 mice per group). (**F**) Sirt1 protein expression levels in the atrial tissues (*n* = 3 mice per group). (**G**) Representative images of Nampt immunohistochemistry in the atrial tissues. Nampt was mainly expressed in the cytoplasm of atrial cardiomyocytes. Scale bar: 50 µm. (**H**) Positively stained areas in the atrial tissues by immunohistochemistry. (**I**) Atrial NAD/NADH ratio in four studied groups (*n* = 4 mice per group). * *p* < 0.05 vs. WT+ND mice. Data are shown as mean ± SD. Statistical comparisons between multiple groups: one-way ANOVA followed by a post-hoc Bonferroni test.

**Figure 2 ijms-21-04655-f002:**
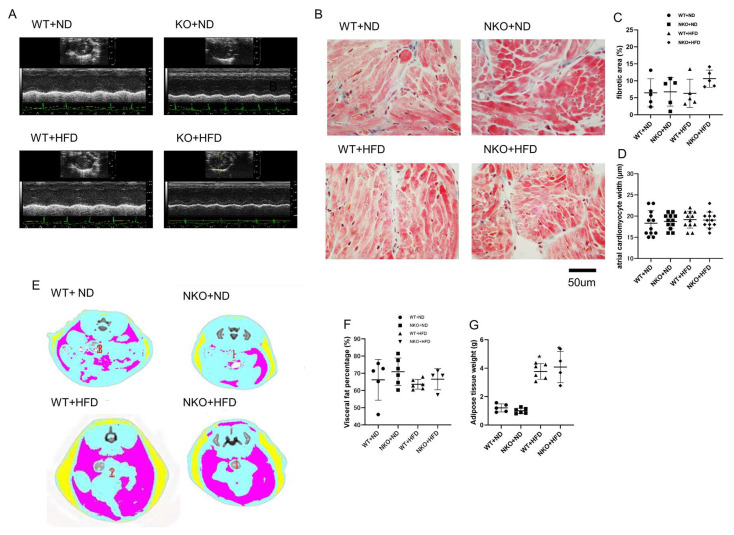
Echocardiography, histology, and adipose tissue measurements. (**A**) Representative M-mode echocardiographic images of a mouse heart. (**B**) Representative images of Masson trichrome staining of the atrial tissues. (**C**) Ratios of fibrotic area to total area (*n* = 5 mice per group). (**D**) calculated atrial cardiomyocytes width (*n* = 12 cells in 3 mice per each group). (**E**) Representative images of abdominal CT scans. Yellow: subcutaneous adipose tissue; pink: intestinal (visceral) adipose tissue. (**F**) Visceral adipose tissue weights measured by abdominal CT scans. (**G**) Subcutaneous adipose tissue weight measured by abdominal CT scan (*n* = 5 mice in WT+ND group, *n* = 6 in NKO+ND group, *n* = 6 in WT+HFD group, *n* = 4 in NKO+HFD group). **p* < 0.05 vs. WT+ND mice. Statistical comparisons between multiple groups: one-way ANOVA followed by a post-hoc Bonferroni test.

**Figure 3 ijms-21-04655-f003:**
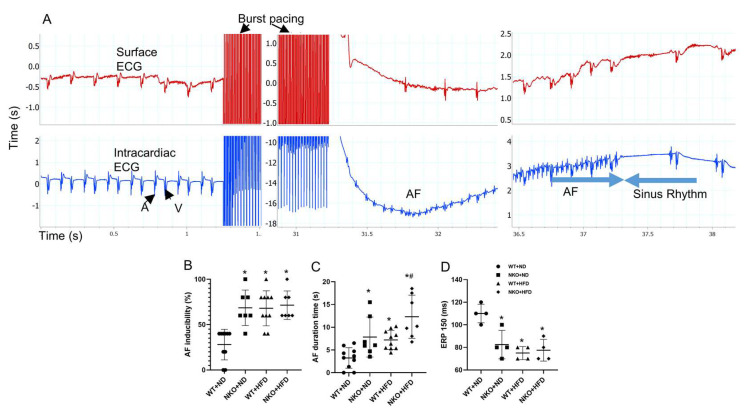
Electrophysiological study. (**A**) Typical mouse surface electrocardiogram (top) and endocardial electrogram (ECG; bottom) recordings during induced AF. A: intra-atrial ECG; V: intra-ventricular ECG. (**B**) scatter chart of AF inducibility. AF inducibility defined as the percentage of AF occurrence in 5 times pacing. (**C**) scatter chart of AF duration time. AF duration defined as the average AF duration time when AF happened. (**D**) AERP at the basic cycle length of 150 ms. * *p* < 0.05 vs. WT+ND mice; ^#^
*p* < 0.05 vs. WT+HFD mice (*n* = 10 in WT+ND group, *n* = 7 in NKO+ND group, *n* = 10 in WT+HFD group, *n* = 7 in NKO+HFD group). Statistical comparisons between multiple groups: one-way ANOVA followed by a post-hoc Bonferroni test.

**Figure 4 ijms-21-04655-f004:**
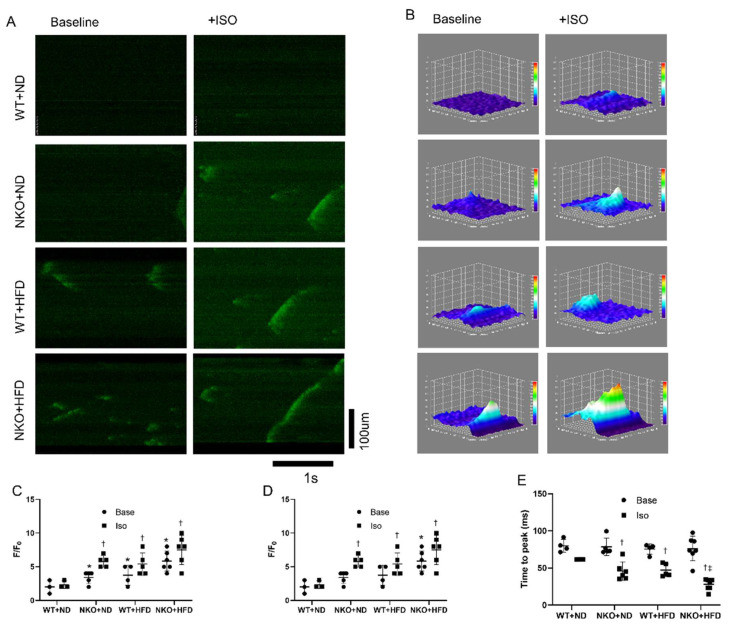
Calcium imaging of isolated cardiomyocytes. (**A**) Representative calcium images of isolated cardiomyocytes at baseline and after application of Iso obtained by confocal microscopy. (**B**) Representative images of 3D surface plot of calcium sparks of each groups. (**C**) Numbers of calcium sparks and mini waves per scan. Mini waves are defined as the calcium waves which did not transport throughout the cardiomyocyte, but only a portion of the cell. Base indicates baseline condition, and Iso indicates condition after adding isoproterenol (Iso). (**D**) Scatter chart of fractional fluorescence increases (F/F_0_). (**E**) Scatter chart of time to peak florescence signal. * *p* < 0.05 vs. WT+ND mice; ^†^
*p* < 0.05 vs. WT+ND+Iso mice; ^‡^
*p* < 0.05 vs. WT+HFD+Iso mice (*n* = 10 in WT+ND group, *n* = 7 in NKO+ND, *n* = 7 in WT+HFD group, *n* = 6 in NKO+HFD group). Statistical comparisons between multiple groups: one-way ANOVA followed by a post-hoc Bonferroni test.

**Figure 5 ijms-21-04655-f005:**
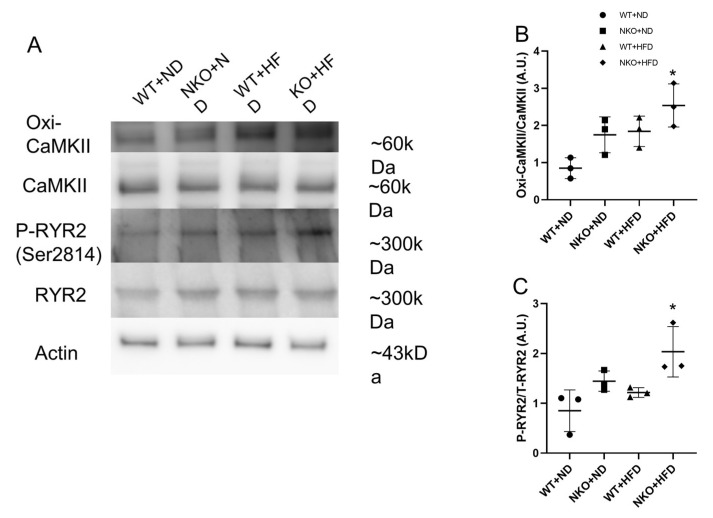
Western blotting of calcium handling pathway-related molecules. (**A**) Representative western blots for proteins with key roles in the calcium handling pathway. β-Actin was assessed as an internal control. (**B**) Ratios of oxidized CaMKII to total CaMKII. (**C**) Ratios of phosphorylated RyR2 to total RyR2. * *p* < 0.05 vs. WT+ND mice (*n* = 3 per each groups). Statistical comparisons between multiple groups: one-way ANOVA followed by a post-hoc Bonferroni test.

**Figure 6 ijms-21-04655-f006:**
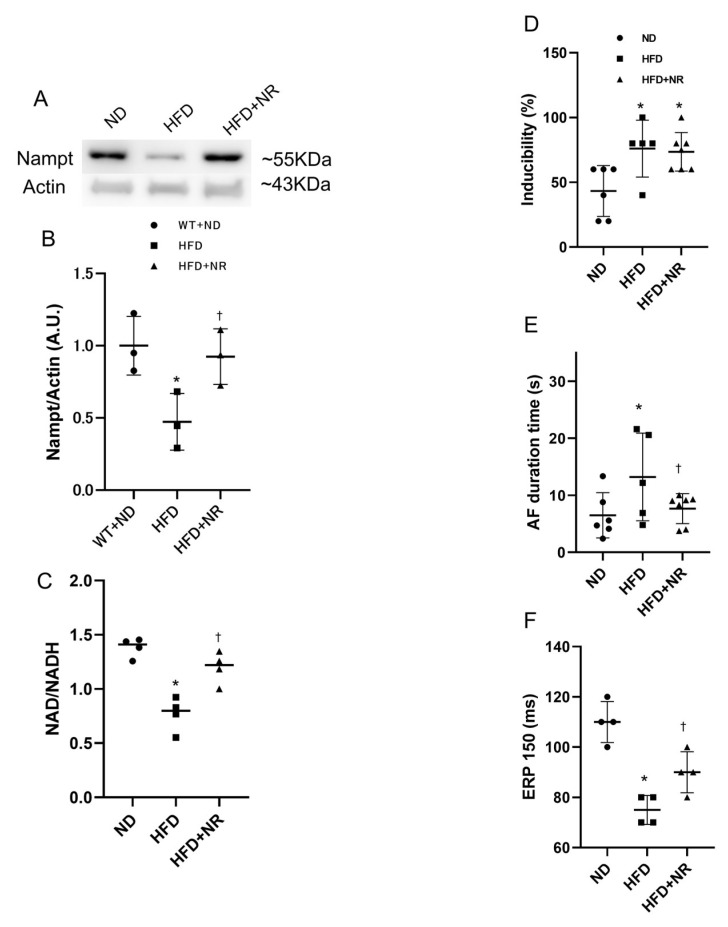
Effect of NR on HFD-induced AF. (**A**) Representative images of western blots for Nampt. (**B**) Nampt protein expression levels in the atrial tissues (*n* = 3 mice per each group). (**C**) Scatter chart of NAD/NADH ratio (*n* = 4 mice per each group). (**D**) Scatter chart of AF inducibility. (**E**) Scatter chart of AF duration. (**F**) Scatter chart of AERP. * *p* < 0.05 vs. ND mice; ^†^
*p* < 0.05 vs. HFD mice (*n* = 6 in ND group, *n* = 5 in HFD group, *n* = 7 in HFD+NR group). Statistical comparisons between multiple groups: one-way ANOVA followed by a post-hoc Bonferroni test.

**Table 1 ijms-21-04655-t001:** Echocardiographic Findings.

Echocardiographic Parameters	WT		NKO	
	ND	HFD	ND	HFD
LVDd (mm)	3.89 ± 0.49	3.71 ± 0.51	3.96 ± 0.47	3.71 ± 0.35
LVDs (mm)	2.50 ± 0.21	2.51 ±0.18	2.49 ± 0.16	2.53 ± 0.18
FS (%)	32.48 ± 9.53	31.66 ± 7.06	31.59 ± 6.40	29.82 ± 5.92
EF (%)	59.66 ± 12.73	60.10 ± 10.45	59.92 ± 9.34	57.41 ± 8.82
LAD (mm)	1.09 ± 0.06	1.10 ± 0.15	1.11 ± 0.02	1.12 ± 0.07
LVSV (μL)	50.28 ± 15.33	51.10 ± 13.19	58.58 ± 21.84	57.28 ± 12.96

LVDd: left ventricular diastolic diameter; LVDs: left ventricular systolic diameter; FS: fraction shortening; EF: ejection fraction; LAD: left atrial diameter; LVSV: left ventricular stroke volume; *p* > 0.05, not significant vs. WT+ND mice.

**Table 2 ijms-21-04655-t002:** Blood Pressure, Body Weight, and Tissue Weight Measurements.

Basic Informations	WT		NKO	
	ND	HFD	ND	HFD
SBP (mmHg)	130.4 ± 11.1	130.6 ± 8.2	129.6 ± 14.5	137.2 ± 12.7 *
DBP (mmHg)	105.4 ± 9.5	97.6 ± 7.7	101.8 ± 11.5	109.8 ± 9.5 *
MBP (mmHg)	113.6 ± 9,6	107 ± 6.7	110.6 ± 11.9	118.4 ± 6.1 *
BW (g)	26.84 ± 0.62	34.04 ± 3.30	26.48 ± 1.07	35.36 ± 1.55 *
HW/BW (mg/g)	4.61 ± 0.28	3.54 ± 1.55 *	4.37 ± 0.20	3.50 ± 0.32 *
LV/BW (mg/g)	3.21 ± 0.16	2.96 ± 0.78 *	3.12 ± 0.17	3.02 ± 0.24 *
AV/BW (mg/g)	0.32 ±0.05	0.29 ± 0.10	0.30 ±0.06	0.30 ± 0.07

SBP: systolic blood pressure; DBP: diastolic blood pressure; MBP: mean blood pressure; BW: body weight; HW/BW: heart weight/body weight ratio; LV/BW: left ventricular weight/body weight ratio; AV/BW: atrial weight/body weight ratio; * *p* < 0.05 vs. WT+ND mice.

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
