# Peer review of "Nicotinamide Phosphoribosyltransferase (Nampt)/Nicotinamide Adenine Dinucleotide (NAD) Axis Suppresses Atrial Fibrillation by Modulating the Calcium Handling Pathway"

_ijms, 2020, doi:10.3390/ijms21134655_

Round 1

Reviewer 1 Report

This paper presented by Feng et al is an interesting study on the role of the enzyme Nampt in atrial fibrillation. The authors use both genetically modified mice and a pharmacological approach to test their hypothesis. It is a well-done work, and in general the conclusions are supported by the results obtained.

I have some observations to make to the authors:

1.Since the conclusions mention: “…the present findings suggest that HFD and NKO-induced NADPH degradation increased ROS production, which directly enhanced oxidation of CaMKII and phosphorylation of RyR2, thereby augmenting AF occurrence...”  the increase in ROS production in NKO should be confirmed.

Minor points:

  1. Abstract needs revision. Line 27” … and the calcium…”. Line 29: Overall, this work demonstrates that both the partially deletion of Nampt facilitated HFD-induced AF…
  2. The number of animals in the graphs does not always correspond to the number in the legend. Check.
  3. The graph in Figure 2E is repeated (2C). Correct and add the corresponding graph.
  4. The description of the NR treatment is missing.
  5. The list of primers used does not correspond to the mRNAs analyzed in the paper. Check.
  6. Was the histology contrasted with DAPI?

Author Response

We appreciate the reviewer1’s insightful comments, which helped us to improve the manuscript substantially. We believed we have addressed the concerns that were raised by the reviewer. The corresponding changes and improvements made in revised manuscript are summarized in our responses below.

1.since the conclusions mention:“...the present findings suggest that HFD and NKO-induced NADPH degradation increased ROS production, which directly enhanced oxidation of CaMKII and phosphorylation of RyR2, thereby augmenting AF occurrence…” the increase in ROS production in NKO should be confirmed.

We agree with the reviewer’s comment, previous studies have confirmed that the ROS production is affected by Nampt expression. In this study, we had mentioned in the discussion that the ROS production is dependent by the Nampt expression from line 210 to line 213. Also, we raised the issue at the end of the manuscript as part of the limitations from the Lane 257 to 261. We also, added the reference 42 to 44 to prove the increase of ROS production. According to your comment, we tried to confirm the ROS production, even though we have the sample ready to experiment, due to the current unrest caused by Covid-19, and we do not have the ROS kit right now. It would cause us at least one month. Since the editor required the revised manuscript to be uploaded within 5 days, we had to leave it as limitations.

Minor points

  1. Abstract needs revision. Line 27” … and the calcium…”. Line 29: Overall, this work demonstrates that both the partially deletion of Nampt facilitated HFD-induced AF…

Thank reviewer for the comment. As the reviewer pointed out, the grammar of the sentence was incorrect. And we had corrected the first one to” mice and the calcium handling pathway was…”; and the second one to “that partially deletion of Nampt…”. Please see in the Lane 27 and 29 to see the revised version.

  1. The number of animals in the graphs does not always correspond to the number in the legend. Check.

Thank reviewer for the comment. As the reviewer pointed out, the number of qPCR results of the Figure 1A and Figure 1B was incorrect. It is because at first when we analyzed the data, we set each sample for two separate sets when running qPCR. And we didn’t notice it, hence all the qPCR results were wrong, we have revaluated all the row data and calculated again and made the new chart graph in the Figure 1A and 1B.

  1. The graph in Figure 2E is repeated (2C). Correct and add the corresponding graph.

Thank reviewer for the comment. As the reviewer pointed out, the figure 2E was repeated due to the careless reviewed on the first author. We have added the corresponding graph for the figure 2E. Please check it out in the revised manuscript.

  1. The description of the NR treatment is missing.

Thank reviewer for the comment. As the reviewer pointed out, the NR treatment method was missed in the methods section. We have added the necessary information on the NR treatment protocol. The content was put from Lane 376 to Lane 380.

  1. The list of primers used dose not correspond to the mRNAs analyzed in the paper. Check.

Thank reviewer for the comment. As the reviewer pointed out, the list of primers did not correspond to the analysis in the paper. We accidentally put some tested primers that in the end did not use in the final manuscript. We apology for this error. We have erased all the primers that did not come out in the manuscript. Please check the revised manuscript Lane 339 and Lane 340.

  1. Was the histology contrasted with DAPI?

Thank reviewer for the question. Yes, the histology was contrasted with DAPI as showed in Lane 353 “... and mounting on slides with mounting medium containing DAPI…”

We appreciate all your insightful comments and questions. Thank you for taking the time to help us improve this manuscript.

Reviewer 2 Report

In the study “Nicotinamide phosphoribosyltransferase (Nampt)/nicotinamide adenine dinucleotide (NAD) axis suppresses atrial fibrillation by modulating the calcium handling pathway“ Feng et al. analyze the relation of Nampt expression in heterozygous Nampt knockout (NKO) mice under normal and high-fat-diet (HFD) and its correlation with the induction of atrial fibrillation and CaMKII signaling. The authors show that both HFD as well as Nampt heterozygosity leads to downregulation of Nampt mRNA and protein expression (the latter in Western Blots and in immunohistochemical analysis). Cardiac function and body weight is not affected in NKO but adipose tissue weight/volume (see below) assessed by CT scan analysis is supposed to be increased. Neither NKO nor HFD (nor its combination) increases atrial fibrosis but systolic blood pressure. NKO, HFD or its combination all increase the proportion of mice in which atrial fibrillation (AF) can be evoked by burst stimulation. Confocal calcium imaging indicates that under these three conditions the frequency of Ca2+ sparks is increased in cardiomyocytes. NKO plus HFD increases oxidation of CaMKII as well as RYR2 phosphorylation. Interestingly, treatment with the NAD precursor NR (nicotinamide riboside) increases Nampt mRNA and protein expression and reduces AF duration and the atrial effective refectory period (AERP). The authors conclude based on their results that the Nampt/NAD axis is a potent therapeutic target for RF.

This is a very interesting study about this NAD/Nampt pathway and its relevance for atrial arrhythmias. Several issues need to be addressed.

Major issues.

  1. The authors study the expression of Nampt, the key enzyme in the salvage pathway for NAD generation. However, there are alternative pathways to generate NAD and key enzymes such as nicotineamide riboside kinase Nmrk 1,2 as well as NAD synthetase (Nadsyn1) which should also be included in the expression analysis in the different conditions.
  2. As NAD levels are implicated to play a major role for the pathophysiology in this study, the question remains whether NAD levels are lowered as suggested and whether NR treatment leads to correction as studied recently in a heart failure model (Circulation 137: 2256-2273,2018).
  3. In Figure 1A six data points are shown but in the methods section it is indicated that four biological replicates (mice) were studied. Thus, only data points representing biological replicates should be shown and used for statistical analysis. A number of biological replicates should be indicated for every data sets shown in figures and results.
  4. Are Nampt expression levels also changed in the ventricles?
  5. Figure 2A and table 1: The authors state that there is the NKO nor HFD has any effect on cardiac function and morphology. Is cardiomyocyte size (atria) changed?
  6. It seems that atrial fibrosis (figure 2B, and C) may be increased in the NKO plus HFD condition. How many mice would be needed to be studied with the actual effect size of this study to show a difference of e.g. 20 or 30% in fibrosis (Power analysis)?

In addition, in Figure 2 panels C and E seems to be the identical.

  1. It is not clear from the figure legend or methods section, how fat weight is calculated. I also assume that is rather fat volume than weight which can be calculated from the CT scans. How many scans were analyzed to calculate such volume or area? It would be helpful in the legend of Figure2F to mention that the fat tissue corresponds to the magenta (?) color.
  2. Table 2: How was blood pressure measured? The method is lacking. In addition to systolic blood pressure also mean arterial blood pressure and diastolic blood pressure should be reported. Body weight indicated in table 2: is this starting body weight before starting of the diet or at the end of the study? Was starting body weight equal in all groups?
  3. The analyzed parameter for AF susceptibility should be expressed more clearly in the results section. Is that the proportion (%) of mice in which AF could be evoked? Is there any change in the time to AF onset? In figure 3 the quality of the labeling of X and Y axis is very poor. In the intra-cardiac ECG traces, ventricular ECG is barely recognizable and atrial trace not visible. Quality needs to be improved significantly.
  4. Regarding calcium imaging: was calcium imaging performed in ventricular cardiomyocytes? Could it be done in atrial cardiomyocytes, which would be a much better surrogate with the in vivo analysis of atrial fibrillation. In figure 4C, the left and right plots under each conditions should be clearly labeled. Is this without and with Isoproterenol stimulation or is it calcium sparks and mini waves? What are the criteria for mini waves? Details should be indicated more precisely in the figure legend. Can NR treatment reduce the frequency of sparks? Do the differences in sparks translate into alterations of calcium transients during pacing conditions?
  5. In the methods section the description of the mouse model needs to be clearly indicated by the reference describing the generation of the NKO mice. Also. the genetic background of the mice and gender needs to be indicated explicitly. In general, the methods should be better described. For example, the details of NR diet are missing, as well as calcium imaging parameters or the protocol to measure and define calcium sparks; which was the Iso concentration and stimulation procedure.
  6. In the methods section of the RNA expression analysis it should be elaborated in more detail how the quantification of transcripts from the investigated genes was calculated. Was this a densitometric analysis after 40 cycles of amplification? There are also primer pairs indicated for HCN4, KCNJ2, SCN5, CACNA1c etc., but expression levels of these genes has not been indicated in the figures. Was there any quality control of the RNA performed. What means that “the results were obtained from 2-3 independent measurements (n=4 mice per group)”; the samples were pooled and what is plotted are duplicates? See also point6.
  7. Acknowledgments. The authors acknowledged the provider of Nampt knockout mice or “for providing the Nampt knockout mice cell line”… Lane 385/386. Wording should be corrected.
  8. Abstract lines 23 and 27 punctuation.
  9. Please include the description of abbreviations when used for the first time sand in the legend of each figure. Many abbreviations appear only at the end in the method section (i.e. AERP).
  10. Please complete the legend of the figures also mentioning the duration of the HF (also NR as required) diet and abbreviations descriptions.
  11. Line 103: It heterozygous instead of heterogeneous what the authors mean?
  12. Correct the description of Figure 4 in the text regarding the mentioning of the panels. E is not mentioned. In addition, please mention to which signal refers the ratio F/Fo. To the complete spark, to the portion of the line scan and also would be helpful that is an indication of total Ca amount of the measured signal.

Author Response

We appreciated the reviewer’s insightful comments, which have helped us to improve the manuscript substantially. Below is a point-by-point response to the reviewer’s comments and concerns.

  1. The authors study the expression of Nampt, the key enzyme in the salvage pathway for NAD generation. However, there are alternative pathways to generate NAD and key enzymes such as nicotinamide riboside kinase Nmrk 1,2 as well as NAD synthetase (Nadsyn1) which should also be included in the expression analysis in the different conditions.

Thank reviewer for the comments. We agree with the reviewer that not only Nampt control the production of NAD, but also the Nmrk 1,2 and Nadsyn1 as well. We have added the contents in the discussion section from line 242 to line 245. According to your comment, we tried to confirm the different NAD pathway, even though we have the sample ready to experiment, due to the current unrest caused by Covid-19, and we do not have the necessary antibodies right now. It would cause us at least one month. Since the editor required the revised manuscript to be uploaded within 5 days, we had to leave it as limitations. The references were added at No.38 and No.39.

  1. As NAD levels are implicated to play a major role for the pathophysiology in this study, the question remains whether NAD levels are lowered as suggested and whether NR treatment leads to correction as studied recently in a heart failure model (Circulation 137: 2256-2273,2018).

Thank reviewer for the comments. We agreed with the reviewer that NAD levels play an integral role in the pathophysiology in this study. We have added the contents in the discussion section from line 257 to line 261. According to your comment, we tried to confirm the NAD contents, even though we have the sample ready to experiment, due to the current unrest caused by Covid-19, and we do not have the necessary kit right now. It would cause us at least one month. Since the editor required the revised manuscript to be uploaded within 5 days, we had to leave it as limitations. The references were added at No.42-No.44.

  1. In figure 1A six data points are shown but, in the methods section, it is indicated that four biological replicates (mice) were studied. Thus, only data points representing biological replicates should be shown and used for statistical analysis. A number of biological replicants should be indicated for every data sets shown in figures and results.

Thank reviewer for pointing out our errors. It is because, at first, when we analyzed the data, we set each sample for two separate sets when running qPCR. And we didn’t notice it, hence all the qPCR results were wrong, we have revaluated all the row data and calculated again and made the new chart graph in the figure 1A and 1B.

  1. Are Nampt expression levels also changed in the ventricles?

Thank reviewer for the question. Sadly, we did not evaluate the Nampt expression in the ventricles in this study. However, in another study, the systemic Nampt heterozygous knockout mice were shown decrease in Nampt expression level in the ventricles. (American Journal of Physiology 317: 711-725, 2019)

  1. Figure 2A and table1: the authors state that there is the NKO nor HFD has any effect on cardiac function and morphology. Is cardiomyocyte size (atria) changed?

Thank reviewer for the question. We looked back on our MT staining images took by the 40x lens microscopy and used the BZ-X analyzer to calculate the width of the atrial cardiomyocytes. We checked 12 cells from 3 mice in each group and the result showed no significant differences between the four groups (Figure2D). Please check out from the revised manuscript line 88.

  1. It seems that atrial fibrosis (figure 2B, and C) may be increased in the NKO plus HFD condition. How many mice would be needed to be studied with the actual effect size of this study to show a difference of e.g. 20 or 30% in fibrosis.

Thank reviewer for the question. The percentages of fibrosis were low, it was average 6.46% in the WT+ND group compared to 10.63% in the NKO+HFD group. According to the math formula, to achieve a p value < 0.05, the sample size would be bigger than 70 mice per each group because the standard deviations were high. This number is very big, and we think it is almost impossible to achieve.

  1. It is not clear from the figure legend or methods section; how fat weight is calculated. I also assume that is rather fat volume than weight which can be calculated from the CT scans. How many scans were analyzed to calculate such volume or area? It would be helpful in the legend or Figure 2F to mention that the fat tissue corresponds to the magenta (?) color.

Thank reviewer for the comments and questions. The method of weight calculation was according to the previous paper as we cited in the references No. 48. It provided the formula to calculate both visceral and subcutaneous fat tissue weights. The numbers of scans were not the same in every mouse, because each scan was a fit 1-mm slice images. And each mouse is a little bit different in shape and length. We calculated the fat tissue from the proximal end of lumbar vertebra L1 to the distal end of lumbar vertebra L6, as indicated in the methods. Also, in the figure legend of figure 2D, we wrote that the yellow part was subcutaneous adipose tissue and the pink part was visceral adipose tissue.

  1. Table 2: How was blood pressure measured? The method is lacking. In addition to systolic blood pressure also mean arterial blood pressure and diastolic and diastolic blood pressure should be reported. Body weight indicated in table 2: is this starting body weight before starting of the diet or at the end of the study? Was starting body weight equal in all groups?

Thank reviewer for the comments and questions. Sorry that we did not put the method of the blood pressure. We used a non-invasive kit to test mice blood pressure through the tail. The method had been added to the manuscript from line 275 to line 277. We did not put the mean arterial blood pressure and diastolic blood pressure because it followed the same pattern as systolic blood pressure. And here, we added both data to table 2, as the reviewer suggested. The body weight in table 2 was the weight of the mice before sacrifice. All mice’s starting body weight was equal in all groups.

  1. The analyzed parameter for AF susceptibility should be expressed more clearly in the results section. Is that the proportion (%) of mice in which AF could be evoked? Is there any change in the time to AF onset? In figure 3 the quality of the labeling of X and Y axis is very poor. In the intra-cardiac ECG traces, ventricular ECG is barely recognizable and atrial trace not visible. Quality needs to be improved significantly.

Thank reviewer for the comments and questions. The parameter for AF susceptibility is explained in the result section as the percentage of occurrences of the AF within its 5 times of burst pacing. The content can be found in Line 116. The quality of the labeling was rearranged as suggested by the reviewer to improve the figure quality.

  1. Regarding calcium imaging: was calcium imaging performed in ventricular cardiomyocytes? Could it be done in atrial cardiomyocytes, which would be a much better surrogate with the in vivo analysis of atrial fibrillation. In figure 4C, the left and right plot under each condition should be clearly labeled. Is this without and with Isoproterenol stimulation or is it calcium sparks and mini waves? What are the criteria for mini wave? Details should be indicated more precisely in the figure legend. Can NR treatment reduce the frequency of sparks? Do the differences in sparks translate into alterations of calcium transients during pacing condition?

Thank the reviewer for the comments and questions. We did use both left and right atriums to isolate the atrial cardiomyocytes. We rearranged the method to avoid misunderstanding in line 309. The left and right plots under each condition were labeled in Figure 4E. If the reviewer wishes the label to be put on every cart, we will further add the label to avoid the misunderstanding. As for the NR treatment, unfortunately we didn’t have enough NR to perform the calcium imaging. So, we don’t know how NR would do to the calcium handling pathway. Also, the differences in sparks translate into alterations of calcium transients during pacing condition.

  1. In the methods sections the description of the mouse model needs to be clearly indicated by the reference describing the generation of the NKO mice. Also, the genetic background of the mice and gender needs to be indicated explicitly. In general, the methods should be better described. For example, the details of NR diet are missing, as well as calcium imaging parameters or the protocol to measure and define calcium sparks, which was the Iso concentration and stimulation procedure.

Thank reviewer for the comments. We agree with the reviewer that the method was not clearly described, so we added the reference describing the generation of the NKO mice, the genetic background and gender from line 270. The NR treatment was added to the methods in line 373. The calcium imaging methods were also refined, added the calcium imaging parameters and measure and define of calcium sparks as the reviewer commented from line 315 to line 321.

  1. In the methods section of the RNA expression analysis it should be elaborated in mor detail how the quantification of transcripts from the investigated genes was calculated. Was this a densitometric analysis after 40 cycles of amplification? There are also primer pairs indicated for HCN4, KCNJ2, SCN5, CACNA1c etc., but expression levels of these genes have not been indicated in the figures. Was there any quality control of the RNA performed? What means that “the results were obtained from 2-3 independent measurements (n=4 mice per group)”; the samples were pooled and what is plotted are duplicates? See also point 6.

Thank reviewer for the comments and questions. This was a densitometric analysis after 40 cycles of amplification as mentioned in the manuscript. As the reviewer pointed out, the list of primers did not correspond with the analysis in the paper. We accidentally put some tested primers that in the end did not use in the final manuscript. We apology for this error. We have erased all the primers that did not come out in the manuscript, as shown in line 339. For the last question, sorry for the misunderstanding, we just meant that the result was measured multiple times to ensure. We had changed the sentence in order not to cause any misunderstanding from line 336-338.

  1. The authors acknowledged the provider of Nampt knockout mice or “for providing the Nampt knockout mice cell line” … lane 385/386. Wording should be corrected.

Thank reviewer for pointing out our error. We checked out the manuscript and rearranged the contents as indicated in line 405-406.

  1. Abstract lines 23 and 27 punctuation.

Thank reviewer for pointing out our errors. We checked out the manuscript and changed the punctuations accordingly. Please check out the revised manuscript in line 23 and line 27.

  1. Please include the description of abbreviations when used for the first-time sand in the legend of each figure. Many abbreviations appear only at the end in the method section (i.e. AERP).

Thank reviewer for pointing out the errors we made in the paper. We had changed the manuscript as suggested, added all the abbreviations for the first time it occurred. Please check out the revised manuscript from line 60-62, 121-122, 140

  1. Please complete the legend of the figures also mentioning the duration of the HF (also NR as required) diet and abbreviations descriptions.

Thank reviewer for the comment. Sorry for the misunderstanding, we had added the duration of HFD in the revised manuscript. Please check it out in line 60 to 62, as for the duration of NR treatment, we also added to the revised manuscript. Please check it out in line 376 to 380.

  1. Line 103: It heterozygous instead of heterogeneous what the author means?

Thank reviewer for the question. Sorry for the misspelling. Such errors sometimes happene. We are very sorry we didn’t find it out in time. We had rearranged all the misspelling in the manuscript. Please check the revised manuscript from line 104, 109, 198, 270.

  1. Correct the description of Figure 4 in the text regarding the mentioning of the panels. E is not mentioned. In addition, please mention to which signal refers the ratio F/F0. To the complete spark, to the portion of the line scan and also would be helpful that is an indication of total Ca amount of the measured signal.

Thank reviewer for the comments. We had rearranged the Figure 4E figure legends. For the F/F0, Baseline fluorescence (F0) was determined by averaging 10 images without calcium spark activity. Fractional fluorescence increases (F/F0) were determined in areas (2.2x2.2μm) where calcium sparks were detected. The definition was added in the method section. Please check out the revised manuscript from line 315-321.

We appreciate all reviewer’s insightful comments and questions. Thank you for taking the time to help us improve this manuscript.

Reviewer 3 Report

The manuscript presents an interesting investigation regarding the role of Nicotinamide phosphoribosyltransferase (Nampt) and  NAD precursor nicotinamide riboside in modulating atrial fibrillation vulnerability, by using the model of heterozygous Nampt knockout mouse.

The experimental protocol is straightforward; the results are presented in a clear sequence, and adequately discussed.

Some minor points:

Line 64: …to the mRNA expression pattern (Figure 1C, D, E).

Line 103:  “heterogeneous NKO”:  "heterozygous" perhaps. Please check throughout the manuscript the consistency of the term used (line 108, 182, 194, 245)

Line 106: “the ratio of heart weight to BW did not differ significantly among the four groups”: the data from Table 2 appear to indicate that HW/BW significantly differs, if comparing WT+ND vs WT+HFD and vs KNO+HFD.

Line 108: from figure 2E, it seems that no significant differences were found for visceral adipose tissue, since no significance asterisks are presented in the graph.

Line 109: “eirhter fat”: either fat

Line 119: it is better to provide here (rather than at line 285) the meaning of AERP (atrial muscle effective refractory period), since first mentioned here.

Line 137: “under Iso stimulation (Figure 4B)”: Figure 4C

Line 138: “with WT+ND mice (Figure 4C)”: Figure 4D

Line 140: “after Iso stimulation (Figure 4D)” Figure 4E

Lines 194-195: “the differences between heterogeneous conventional knockout mice and adipose tissue-specific homogeneous knockout mice.”: should be “heterozygous” and “homozygous”

Line 233: and (not italics)

Line 244: “a promising therapeutic reagent for cardiovascular diseases”: “reagent” is not a correct term to be used here. Maybe “agent”, or “intervention”

Line 254: “humanely”: probably this term is not appropriate to describe a laboratory procedure. Just it is possible to delete it, since the regulatory rules quoted are enough to explain the correctness of the procedures.

Line 259: “Heterozygous NKO mice”:  Description of more complete characteristics of this strain of KO mice could be appreciated.

Line 345: at room temperature

Line 352: at 4°C for another hour

Figure 2 A: heading for top right panel should be NKO+ND, and for bottom right panel should be NKO+HFD

Figure 3 A: It is not clear the position/meaning of arrow labeled as "A" (intra-atrial) for intracardiac ECG

Author Response

Responses to the comments by Reviewer 3

We thank reviewer 3 for the constructive comments, which have helped us to improve the manuscript.

  1. Line 64: …to the mRNA expression pattern (Figure 1C, D, E).

Thank reviewer for pointing out the error. We had changed the manuscript as requested. Please check out the revised manuscript line 67.

  1. Line 103: “heterogeneous NKO”: “heterozygous” perhaps. Please check throughout the manuscript the consistency of the term used (line 108, 182, 194, 245).

Thank reviewer for pointing out our errors. We have check out the manuscript and rearranged all the misspellings as requested. Please check out the revised manuscript from line 104, 109, 186, 198, 270.

  1. Line 109: “eirhter fat”: either fat.

Thank reviewer for pointing out the spelling error. We had changed to the correct one. Please check out the revised manuscript in line 110.

  1. Line 108: from figure 2E, it seems that no significant differences were found for visceral adipose tissue, since no significance asterisks are presented in the graph.

Thank reviewer for pointing out the error. It was, in fact, no significant differences were found for visceral adipose tissue. We accidentally put the wrong graph for the original figure 2E. Please check out the revised version to see the new figure.

  1. Line 119: it is better to provide here (rather than at line 285) the meaning of AERP (atrial muscle effective refractory period), since first mentioned here.

Thank reviewer for pointing out our error. We have check out the manuscript and changed all the abbreviations at first sights. Please check out the revised manuscript in line 121.

  1. Line 137:” under Iso stimulation (Figure 4B)”: Figure 4C.

Thank reviewer for pointing out the error for the mislabeling the Figure. We had rearranged the label. Please check out the revised manuscript in line 140

  1. Line 138:” with WT+ND mice (Figure 4C)”: Figure 4D.

Thank reviewer for pointing out the error for the mislabeling the Figure. We had rearranged the label. Please check out the revised manuscript in line 141,

  1. Line 140:” after Iso stimulation (Figure 4D)” Figure 4E.

Thank reviewer for pointing out the error for the mislabeling the Figure. We had rearranged the label. Please check out the revised manuscript in line 143.

  1. Line 194-195: “the differences between heterogeneous conventional knockout mice and adipose tissue-specific homogeneous knockout mice.”: should be “heterozygous” and “homozygous”.

Thank reviewer for pointing out error. We had rearranged the misspelling in the manuscript. Please check out the revised manuscript from line 198, 199

  1. Line 233: and (not italics).

Thank reviewer for pointing out the error. We had changed the words for not been italics. Please check out the revised manuscript in line 237.

  1. Line 244: “a promising therapeutic reagent for cardiovascular diseases”: “reagent” is not a correct term to be used here. Maybe “agent” or “intervention”.

Thank reviewer for pointing out the error. We had changed the old word to the correct one. Please check out the manuscript for revised version on line 252.

  1. Line 254: “humanely”; probably this term is not appropriate to describe a laboratory procedure. Just it is possible to delete it, since the regulatory rules quoted are enough to explain the correctness of the procedures.

Thank reviewer for pointing out the error. We erased the word in order to prevent misunderstanding. Please check out the revised manuscript in line 265.

  1. Line 259: “heterozygous NKO mice”: Description of more complete characteristics of this strain of KO mice could be appreciated.

Thank reviewer for the comments. We agree with the reviewer that the method was not clearly described, so we added the reference describing the generation of the NKO mice, the genetic background and gender from line 270, 271.

  1. Line 345: at room temperature.

Thank reviewer for pointing out the error. We had changed the manuscript as reviewer suggested. Please check out the revised manuscript in line 362.

  1. Line 352: at 4°C for another hour.

Thank reviewer for pointing out the error. We had changed the manuscript as reviewer suggested. Please check out the revised manuscript in line 369.

  1. Figure 2A: heading for top right panel should be NKO+ND, and for bottom right panel should be NKO+HFD.

Thank reviewer for pointing out the error. We had rearranged the figure panel names, please check out the revised manuscript.

  1. Figure 3A: It is not clear the position/meaning of arrow labeled as “A” (intra-atrial) for intracardiac ECG

Thank reviewer for the comment. We agree with the reviewer on no clear of the position, so we rearranged the figure 3A, and the meaning we rearranged it in the figure legends. Please check out the revised manuscript in the method section.

We appreciate all of the reviewer’s insightful comments. Thanks for taking the time to help us improve this manuscript.

Round 2

Reviewer 2 Report

Author response to report 1:

Author's Notes

We appreciated the reviewer’s insightful comments, which have helped us to improve the manuscript substantially. Below is a point-by-point response to the reviewer’s comments and concerns.

  1. The authors study the expression of Nampt, the key enzyme in the salvage pathway for NAD generation. However, there are alternative pathways to generate NAD and key enzymes such as nicotinamide riboside kinase Nmrk 1,2 as well as NAD synthetase (Nadsyn1) which should also be included in the expression analysis in the different conditions.

Thank reviewer for the comments. We agree with the reviewer that not only Nampt control the production of NAD, but also the Nmrk 1,2 and Nadsyn1 as well. We have added the contents in the discussion section from line 242 to line 245. According to your comment, we tried to confirm the different NAD pathway, even though we have the sample ready to experiment, due to the current unrest caused by Covid-19, and we do not have the necessary antibodies right now. It would cause us at least one month. Since the editor required the revised manuscript to be uploaded within 5 days, we had to leave it as limitations. The references were added at No.38 and No.39.

The editor show give the authors more time to perform the suggested experiments and the qPCR analysis for Nmrk 1,2 and Nadsyn1 as suggested by this reviewer.

  1. As NAD levels are implicated to play a major role for the pathophysiology in this study, the question remains whether NAD levels are lowered as suggested and whether NR treatment leads to correction as studied recently in a heart failure model (Circulation 137: 2256-2273,2018).

Thank reviewer for the comments. We agreed with the reviewer that NAD levels play an integral role in the pathophysiology in this study. We have added the contents in the discussion section from line 257 to line 261. According to your comment, we tried to confirm the NAD contents, even though we have the sample ready to experiment, due to the current unrest caused by Covid-19, and we do not have the necessary kit right now. It would cause us at least one month. Since the editor required the revised manuscript to be uploaded within 5 days, we had to leave it as limitations. The references were added at No.42-No.44.

The editor show give the authors more time to perform measurements of NAD levels as suggested by this reviewer.

  1. In figure 1A six data points are shown but, in the methods section, it is indicated that four biological replicates (mice) were studied. Thus, only data points representing biological replicates should be shown and used for statistical analysis. A number of biological replicants should be indicated for every data sets shown in figures and results.

Thank reviewer for pointing out our errors. It is because, at first, when we analyzed the data, we set each sample for two separate sets when running qPCR. And we didn’t notice it, hence all the qPCR results were wrong, we have revaluated all the row data and calculated again and made the new chart graph in the figure 1A and 1B.

Issue is sufficiently addressed.

  1. Are Nampt expression levels also changed in the ventricles?

Thank reviewer for the question. Sadly, we did not evaluate the Nampt expression in the ventricles in this study. However, in another study, the systemic Nampt heterozygous knockout mice were shown decrease in Nampt expression level in the ventricles. (American Journal of Physiology 317: 711-725, 2019)

Issue is sufficiently addressed.

  1. Figure 2A and table1: the authors state that there is the NKO nor HFD has any effect on cardiac function and morphology. Is cardiomyocyte size (atria) changed?

Thank reviewer for the question. We looked back on our MT staining images took by the 40x lens microscopy and used the BZ-X analyzer to calculate the width of the atrial cardiomyocytes. We checked 12 cells from 3 mice in each group and the result showed no significant differences between the four groups (Figure2D). Please check out from the revised manuscript line 88.

Issue is sufficiently addressed.

  1. It seems that atrial fibrosis (figure 2B, and C) may be increased in the NKO plus HFD condition. How many mice would be needed to be studied with the actual effect size of this study to show a difference of e.g. 20 or 30% in fibrosis.

Thank reviewer for the question. The percentages of fibrosis were low, it was average 6.46% in the WT+ND group compared to 10.63% in the NKO+HFD group. According to the math formula, to achieve a p value < 0.05, the sample size would be bigger than 70 mice per each group because the standard deviations were high. This number is very big, and we think it is almost impossible to achieve.

I understand the problem, but the the sentence should be changed to „The results showed no significant difference in fibrotic area and atrial cardiomyocytes width among the four experimental conditions with a group size of 5 mice (Figure 2C, 2D)“ or similarly like this to make the limitation of the conclusion transparent since one might obtain a difference in fibrosis when analysing a sufficiently large number of samples.

  1. It is not clear from the figure legend or methods section; how fat weight is calculated. I also assume that is rather fat volume than weight which can be calculated from the CT scans. How many scans were analyzed to calculate such volume or area? It would be helpful in the legend or Figure 2F to mention that the fat tissue corresponds to the magenta (?) color.

Thank reviewer for the comments and questions. The method of weight calculation was according to the previous paper as we cited in the references No. 48. It provided the formula to calculate both visceral and subcutaneous fat tissue weights. The numbers of scans were not the same in every mouse, because each scan was a fit 1-mm slice images. And each mouse is a little bit different in shape and length. We calculated the fat tissue from the proximal end of lumbar vertebra L1 to the distal end of lumbar vertebra L6, as indicated in the methods. Also, in the figure legend of figure 2D, we wrote that the yellow part was subcutaneous adipose tissue and the pink part was visceral adipose tissue.

Issue is sufficiently addressed.

  1. Table 2: How was blood pressure measured? The method is lacking. In addition to systolic blood pressure also mean arterial blood pressure and diastolic and diastolic blood pressure should be reported. Body weight indicated in table 2: is this starting body weight before starting of the diet or at the end of the study? Was starting body weight equal in all groups?

Thank reviewer for the comments and questions. Sorry that we did not put the method of the blood pressure. We used a non-invasive kit to test mice blood pressure through the tail. The method had been added to the manuscript from line 275 to line 277. We did not put the mean arterial blood pressure and diastolic blood pressure because it followed the same pattern as systolic blood pressure. And here, we added both data to table 2, as the reviewer suggested. The body weight in table 2 was the weight of the mice before sacrifice. All mice’s starting body weight was equal in all groups.

Issue is sufficiently addressed.

  1. The analyzed parameter for AF susceptibility should be expressed more clearly in the results section. Is that the proportion (%) of mice in which AF could be evoked? Is there any change in the time to AF onset? In figure 3 the quality of the labeling of X and Y axis is very poor. In the intra-cardiac ECG traces, ventricular ECG is barely recognizable and atrial trace not visible. Quality needs to be improved significantly.

Thank reviewer for the comments and questions. The parameter for AF susceptibility is explained in the result section as the percentage of occurrences of the AF within its 5 times of burst pacing. The content can be found in Line 116. The quality of the labeling was rearranged as suggested by the reviewer to improve the figure quality.

Issue is sufficiently addressed.

  1. Regarding calcium imaging: was calcium imaging performed in ventricular cardiomyocytes? Could it be done in atrial cardiomyocytes, which would be a much better surrogate with the in vivo analysis of atrial fibrillation. In figure 4C, the left and right plot under each condition should be clearly labeled. Is this without and with Isoproterenol stimulation or is it calcium sparks and mini waves? What are the criteria for mini wave? Details should be indicated more precisely in the figure legend. Can NR treatment reduce the frequency of sparks? Do the differences in sparks translate into alterations of calcium transients during pacing condition?

Thank the reviewer for the comments and questions. We did use both left and right atriums to isolate the atrial cardiomyocytes. We rearranged the method to avoid misunderstanding in line 309. The left and right plots under each condition were labeled in Figure 4E. If the reviewer wishes the label to be put on every cart, we will further add the label to avoid the misunderstanding. As for the NR treatment, unfortunately we didn’t have enough NR to perform the calcium imaging. So, we don’t know how NR would do to the calcium handling pathway. Also, the differences in sparks translate into alterations of calcium transients during pacing condition.

Issue is improved but still not sufficiently addressed. In Fig 4 C, D and E there are always TWO scatter charts for each experiment groups. From the figure legend it does not become clear what is analysed in each of the two different scatter plots.

Still spelling errors in figure legend.

Any comment on NR treatment on Ca imaging readouts?

  1. In the methods sections the description of the mouse model needs to be clearly indicated by the reference describing the generation of the NKO mice. Also, the genetic background of the mice and gender needs to be indicated explicitly. In general, the methods should be better described. For example, the details of NR diet are missing, as well as calcium imaging parameters or the protocol to measure and define calcium sparks, which was the Iso concentration and stimulation procedure.

Thank reviewer for the comments. We agree with the reviewer that the method was not clearly described, so we added the reference describing the generation of the NKO mice, the genetic background and gender from line 270. The NR treatment was added to the methods in line 373. The calcium imaging methods were also refined, added the calcium imaging parameters and measure and define of calcium sparks as the reviewer commented from line 315 to line 321.

Issue is sufficiently addressed.

  1. In the methods section of the RNA expression analysis it should be elaborated in mor detail how the quantification of transcripts from the investigated genes was calculated. Was this a densitometric analysis after 40 cycles of amplification? There are also primer pairs indicated for HCN4, KCNJ2, SCN5, CACNA1c etc., but expression levels of these genes have not been indicated in the figures. Was there any quality control of the RNA performed? What means that “the results were obtained from 2-3 independent measurements (n=4 mice per group)”; the samples were pooled and what is plotted are duplicates? See also point 6.

Thank reviewer for the comments and questions. This was a densitometric analysis after 40 cycles of amplification as mentioned in the manuscript. As the reviewer pointed out, the list of primers did not correspond with the analysis in the paper. We accidentally put some tested primers that in the end did not use in the final manuscript. We apology for this error. We have erased all the primers that did not come out in the manuscript, as shown in line 339. For the last question, sorry for the misunderstanding, we just meant that the result was measured multiple times to ensure. We had changed the sentence in order not to cause any misunderstanding from line 336-338.

With 40 cycles and densitometric analysis the method is not „quantitative“ since it is difficult to exclude whether individual probes are already in saturation, primer efficiency not assessed etc.

The results obtained by this endpoint measurement only allow to state whether indvidual mRNA transcripts were detected or not, statements of quantitative comparisons need to be removed from the ms or bona fide qPCR analysis need to be-reformed according to MIQE Guidelines (Minimum Information for Publication of Quantitative Real-Time PCR Experiments; for example, look Bustin et al. 2009. DOI: 10.1373/clinchem.2008.112797)

  1. The authors acknowledged the provider of Nampt knockout mice or “for providing the Nampt knockout mice cell line” … lane 385/386. Wording should be corrected.

Thank reviewer for pointing out our error. We checked out the manuscript and rearranged the contents as indicated in line 405-406.

Issue is sufficiently addressed.

  1. Abstract lines 23 and 27 punctuation.

Thank reviewer for pointing out our errors. We checked out the manuscript and changed the punctuations accordingly. Please check out the revised manuscript in line 23 and line 27.

Issue is sufficiently addressed.

Author Response

To Reviewer #2

We appreciated the reviewer’s insightful comments, which have helped us to improve the manuscript substantially. A point-by-point response to the reviewer’s comments and concerns is expressed as follows.

  1. The authors study the expression of Nampt, the key enzyme in the salvage pathway for NAD generation. However, there are alternative pathways to generate NAD and key enzymes such as nicotinamide riboside kinase Nmrk 1,2 as well as NAD synthetase (Nadsyn1) which should also be included in the expression analysis in the different conditions.

Thank you for the comments. We agree with the reviewer that not only Nampt control the production of NAD, but also the Nmrk 1,2 and Nadsyn1 as well. We have added the contents in the discussion section from line 242 to line 245. According to your comment, we tried to confirm the different NAD pathway, even though we have the sample ready to experiment, due to the current unrest caused by Covid-19, and we do not have the necessary antibodies right now. It would cause us at least one month. Since the editor required the revised manuscript to be uploaded within 5 days, we had to leave it as limitations. The references were added at No.38 and No.39.

The editor show give the authors more time to perform the suggested experiments and the qPCR analysis for Nmrk 1,2 and Nadsyn1 as suggested by this reviewer.

Thank you for giving us more time to improve our paper. We conducted the qPCR analysis for Nmrk1, Nmrk2, and Nadsyn1 as suggested by reviewer. The results showed no significant differences of Nmrk1 and Nadsyn1 in the four studied groups. However, the expression level of Nmrk2 was significantly higher in NKO+HFD group than WT+ND group, hence we added the result to the figure 1C. Please check out the revised manuscript (line 66 to 69, line 83 to 84, line 258 to 259, and line 360 to 361) and figure 1C.

  1. As NAD levels are implicated to play a major role for the pathophysiology in this study, the question remains whether NAD levels are lowered as suggested and whether NR treatment leads to correction as studied recently in a heart failure model (Circulation 137: 2256-2273,2018).

Thank reviewer for the comments. We agreed with the reviewer that NAD levels play an integral role in the pathophysiology in this study. We have added the contents in the discussion section from line 257 to line 261. According to your comment, we tried to confirm the NAD contents, even though we have the sample ready to experiment, due to the current unrest caused by Covid-19, and we do not have the necessary kit right now. It would cause us at least one month. Since the editor required the revised manuscript to be uploaded within 5 days, we had to leave it as limitations. The references were added at No.42-No.44.

The editor show give the authors more time to perform measurements of NAD levels as suggested by this reviewer.

Thank you for giving us more time to perform the experiment. We measured the NAD/NADH by using NAD/NADH quantification kit (Sigma-Aldrich). There was a significant decrease of NAD/NADH ratio in other three groups compared to the WT+ND group. In addition, HFD-induced decrease in the NAD/NADH ratio was partially but significantly recovered by NR treatment, as shown in Figure 6C. Please check out the revised manuscript (line 74 to 76, line 89, line 187 to 188, line 194 to 195, line 208, line 264, and line 397 to 403) and figure 1I and figure 6C.

  1. It seems that atrial fibrosis (figure 2B, and C) may be increased in the NKO plus HFD condition. How many mice would be needed to be studied with the actual effect size of this study to show a difference of e.g. 20 or 30% in fibrosis.

Thank reviewer for the question. The percentages of fibrosis were low, it was average 6.46% in the WT+ND group compared to 10.63% in the NKO+HFD group. According to the math formula, to achieve a p value < 0.05, the sample size would be bigger than 70 mice per each group because the standard deviations were high. This number is very big, and we think it is almost impossible to achieve.

I understand the problem, but the sentence should be changed to „The results showed no significant difference in fibrotic area and atrial cardiomyocytes width among the four experimental conditions with a group size of 5 mice (Figure 2C, 2D)“ or similarly like this to make the limitation of the conclusion transparent since one might obtain a difference in fibrosis when analysing a sufficiently large number of samples.

Thank you for the suggestion. We agree with reviewer’s comments. We changed the sentence as the reviewer suggested. Please check out the revised manuscript form line 96 to 98.

  1. Regarding calcium imaging: was calcium imaging performed in ventricular cardiomyocytes? Could it be done in atrial cardiomyocytes, which would be a much better surrogate with the in vivo analysis of atrial fibrillation. In figure 4C, the left and right plot under each condition should be clearly labeled. Is this without and with Isoproterenol stimulation or is it calcium sparks and mini waves? What are the criteria for mini wave? Details should be indicated more precisely in the figure legend. Can NR treatment reduce the frequency of sparks? Do the differences in sparks translate into alterations of calcium transients during pacing condition?

Thank the reviewer for the comments and questions. We did use both left and right atriums to isolate the atrial cardiomyocytes. We rearranged the method to avoid misunderstanding in line 309. The left and right plots under each condition were labeled in Figure 4E. If the reviewer wishes the label to be put on every cart, we will further add the label to avoid the misunderstanding. As for the NR treatment, unfortunately we didn’t have enough NR to perform the calcium imaging. So, we don’t know how NR would do to the calcium handling pathway. Also, the differences in sparks translate into alterations of calcium transients during pacing condition.

Issue is improved but still not sufficiently addressed. In Fig 4 C, D and E there are always TWO scatter charts for each experiment groups. From the figure legend it does not become clear what is analysed in each of the two different scatter plots.

Still spelling errors in figure legend.

Any comment on NR treatment on Ca imaging readouts?

Thank you for the further comments. We agree with the reviewer’s comments. So, we rearranged the figure4 C, D, E and figure legend. The spelling errors have all resolved. Because we didn’t have data on Ca images after NR treatment, we raised this issues in the discussion section. Please check the revised manuscript from line 147 to 149, line 163 to 164, and line 265 to 267.

  1. In the methods section of the RNA expression analysis it should be elaborated in mor detail how the quantification of transcripts from the investigated genes was calculated. Was this a densitometric analysis after 40 cycles of amplification? There are also primer pairs indicated for HCN4, KCNJ2, SCN5, CACNA1c etc., but expression levels of these genes have not been indicated in the figures. Was there any quality control of the RNA performed? What means that “the results were obtained from 2-3 independent measurements (n=4 mice per group)”; the samples were pooled and what is plotted are duplicates? See also point 6.

Thank reviewer for the comments and questions. This was a densitometric analysis after 40 cycles of amplification as mentioned in the manuscript. As the reviewer pointed out, the list of primers did not correspond with the analysis in the paper. We accidentally put some tested primers that in the end did not use in the final manuscript. We apology for this error. We have erased all the primers that did not come out in the manuscript, as shown in line 339. For the last question, sorry for the misunderstanding, we just meant that the result was measured multiple times to ensure. We had changed the sentence in order not to cause any misunderstanding from line 336-338.

With 40 cycles and densitometric analysis the method is not „quantitative“ since it is difficult to exclude whether individual probes are already in saturation, primer efficiency not assessed etc.

The results obtained by this endpoint measurement only allow to state whether indvidual mRNA transcripts were detected or not, statements of quantitative comparisons need to be removed from the ms or bona fide qPCR analysis need to be-reformed according to MIQE Guidelines (Minimum Information for Publication of Quantitative Real-Time PCR Experiments; for example, look Bustin et al. 2009. DOI: 10.1373/clinchem.2008.112797)

Thank you for pointing out our errors. We are very sorry for the mistake. We performed qPCR using Applied Biosystems 7500 Real-Time PCR System or QuantStudio 5 (ThermoFisher), but not a densitometric analysis. We revised the qPCR experiment methods in more detail. Please check out the revised manuscript from line 348 to 349 and line 354 to 357.

Round 3

Reviewer 2 Report

My concern are now sufficiently addressed, the paper can be published.